# A Forensic Detection Method for Hallucinogenic Mushrooms via High-Resolution Melting (HRM) Analysis

**DOI:** 10.3390/genes12020199

**Published:** 2021-01-29

**Authors:** Xiaochun Zhang, Huan Yu, Qi Yang, Ziwei Wang, Ruocheng Xia, Chong Chen, Yiling Qu, Rui Tan, Yan Shi, Ping Xiang, Suhua Zhang, Chengtao Li

**Affiliations:** 1Department of Forensic Science, Medical School of Soochow University, Suzhou 215123, China; zhangxc19941208@163.com (X.Z.); mldadadidi10@163.com (H.Y.); myjasmine77@163.com (Q.Y.); viviwangzw@163.com (Z.W.); moeqyl77@163.com (Y.Q.); jingbaoer4@163.com (R.T.); 2Shanghai Key Laboratory of Forensic Medicine, Shanghai Forensic Service Platform, Academy of Forensic Science, Ministry of Justice, Shanghai 200063, China; xiaruocheng315@163.com (R.X.); 18883368974@163.com (C.C.); shiy@ssfjd.cn (Y.S.); xiangping2630@163.com (P.X.); 3Department of Forensic Medicine, School of Basic Medical Science, Wenzhou Medical University, Wenzhou 325035, China; 4Health Science Center, College of Medicine and Forensics, Xi’an Jiaotong University, Xi’an 710061, China

**Keywords:** polymerase chain reaction (PCR), high-resolution melting (HRM), DNA barcoding, internal transcribed spacer (ITS), *Psilocybe cubensis*

## Abstract

In recent years, trafficking and abuse of hallucinogenic mushrooms have become a serious social problem. It is therefore imperative to identify hallucinogenic mushrooms of the genus *Psilocybe* for national drug control legislation. An internal transcribed spacer (ITS) is a DNA barcoding tool utilized for species identification. Many methods have been used to discriminate the ITS region, but they are often limited by having a low resolution. In this study, we sought to analyze the ITS and its fragments, ITS1 and ITS2, by using high-resolution melting (HRM) analysis, which is a rapid and sensitive method for evaluating sequence variation within PCR amplicons. The ITS HRM assay was tested for specificity, reproducibility, sensitivity, and the capacity to analyze mixture samples. It was shown that the melting temperatures of the ITS, ITS1, and ITS2 of *Psilocybe cubensis* were 83.72 ± 0.01, 80.98 ± 0.06, and 83.46 ± 0.08 °C, and for other species, we also obtained species-specific results. Finally, we performed ITS sequencing to validate the presumptive taxonomic identity of our samples, and the sequencing output significantly supported our HRM data. Taken together, these results indicate that the HRM method can quickly distinguish the DNA barcoding of *Psilocybe cubensis* and other fungi, which can be utilized for drug trafficking cases and forensic science.

## 1. Introduction

There are hundreds of hallucinogenic mushrooms found in nature that contain active compounds, and most of them can be abused and hazardous to human health [1]. In recent years, the consumption of natural biological products with hallucinogenic properties, such as *Psilocybe cubensis*, has greatly increased [2,3]. The abuse of hallucinogenic mushrooms, plants, or other new types of narcotics leads to a negative influence on society, while the existence of many similar species, various hallucinogenic components, and widespread distribution all make the species identification of these hallucinogens difficult [4]. Many countries have varying regulation levels or bans on hallucinogenic mushrooms, such as the United States (US), the United Kingdom (UK), Canada, and China [5]. Therefore, it is imperative to identify hallucinogenic mushrooms or plants related to criminal proceedings, such as drug-related deaths and drug trafficking, which is also of great significance in forensic investigations.

At present, the detection of hallucinogenic mushrooms and plants primarily relies on methods ranging from morphological and histological characteristics, gas chromatography-mass spectrometry (GC-MS), high-performance liquid chromatography (HPLC), and other toxicological examinations [6,7,8]. However, the morphological characteristics are obscured on dried or powdered samples, and toxicological methods hardly function in degraded samples due to a decreased chemical composition [9]. As a supplement to chemical analysis, a DNA-based approach has been proven to be stable over time. Several lines of studies have used DNA sequences for inference in cradle and hallucinogenic mushrooms or plants species with a phylogenetic relationship [10,11,12,13]. The DNA barcoding assay is also a genetic approach that discriminated one or a few specific DNA regions for species identification. Among these specific regions, the internal transcribed spacer (ITS1-5.8S-ITS2, ITS), flanked by the 18S and 28S rRNA genes, is highly polymorphic between species [12,13,14]. The study of Lee et al. suggested that the genera *Psilocybe* and *Panaeolus* could be identified on the basis of ITS1 sequence differences [15]. The Fungal Barcode Consortium was proposed for the ITS region as the universal DNA barcoding for fungal species based on the level of sequence variability and utility in discriminating species [16]. Considering the reliability of the ITS region, this program merits attention for examining the ITS and establishing a database of hallucinogenic mushrooms and plants based on DNA barcoding, which would allow researchers to share species information easily and identify species accurately.

In molecular systematics, ITS sequencing is the most common method for mushroom identification. However, analysis of ITS sequencing takes too much time due to the high requirements of the precise technical operation and the variety of instruments. Nuclear DNA sequencing is also not commonly used in many forensic laboratories. Apart from sequencing, fluorescent random amplified microsatellites (F-RAMS) is also a supplementary method to identify mushrooms, which has been verified as a reliable method that can cluster genetically close species together [17,18]. Nevertheless, F-RAMS demands multiple samples from the same species to obtain a robust proof of principle for mushroom species taxonomic studies, which is certainly complex and time-consuming.

The technology of high-resolution melting (HRM) is a novel and developing post-PCR method for generating DNA barcoding data, which has been extensively used in many forensic laboratories [19,20,21]. The HRM melting curve has prominent capacity to discriminate DNA barcodes based on their composition, length, or GC content, which allows for the examination of genetic variations within DNA sequences. Moreover, it supports forensic evidence of the suspicious species [19,20,21,22,23]. For example, it has been demonstrated that HRM could identify *Cannabis sativa* and “legal high” drugs, among other species [21]. A pioneer study also proposed a PCR HRM duplex assay that can simultaneously detect *Psilocybe cubensis* and *Cannabis sativa* DNA [23]. In comparison to other discriminatory methods, the HRM analysis is a rapid, user-friendly, cost-effective, and highly reproducible approach that can accurately screen a large number of samples for the presence of specific targets, including the variations within the ITS region. In this paper, we suggested the HRM assay to evaluate the specificity of ITS, ITS1, and ITS2 amplicons from 36 mushroom species, including *Psilocybe cubensis* and other related species. The discriminative performance of HRM was assessed based on reproductivity, sensitivity, and a mixture test. Sanger sequencing was also conducted to validate the presumptive taxonomic identity of our samples. Our study aimed to provide a data basis for the preliminary construction of a DNA barcoding database of hallucinogenic mushrooms and plants, investigating a rapid, scientific, and accurate method for discriminating mushroom species.

## 2. Materials and Methods

### 2.1. Samples

Samples of *Psilocybe cubensis*, *Cannabis sativa*, and other hallucinogenic toxic mushrooms were gathered from forensic laboratories. Other edible mushrooms were collected from Yunnan, Hunan, Guangxi, Shanghai, and other places in China, where there are abundant mushroom resources, meaning that local residents often eat hallucinogenic mushrooms by mistake. All samples were previously identified based on morphological characteristics. The material numbers and detailed descriptions of our samples are summarized in Appendix A.

### 2.2. Genomic DNA Extraction

The genomic DNA of samples was extracted using the DNeasy Plant Pro Kit according to the manufacturer’s protocol (Qiagen, Hilden, Germany). All templates were quantified with a NanoDrop 2000 spectrophotometer (Thermo Fisher Scientific, Waltham, MA, USA) and finally diluted to 1.0 ng/µL for PCR amplification and HRM analysis.

### 2.3. Primer Selection

The primers are listed in Table 1, which were all previously designed by White et al. [24]. As described in their study, universal primers ITS4 and ITS5 were used to amplify the whole ITS sequence (ITS1-5.8S-ITS2) of *Psilocybe cubensis*, *Cannabis sativa*, and other mushrooms. The primers ITS2 and ITS5 were selected to amplify the ITS1 sequence, and primers ITS4 and ITS86 were selected to amplify the ITS2 sequence. Primers were synthesized by Sangon Biotech (Shanghai, China), quantified using a NanoDrop 2000 spectrophotometer, and diluted to 10 µM.

### 2.4. Specificity Studies with HRM Analysis

The ITS, ITS1, and ITS2 regions were tested in triplicate via real-time PCR HRM assay in all samples. A type-it HRM PCR Kit (Qiagen, Mainz, Germany) was used in a real-time PCR reaction. PCR amplification, DNA melting, and fluorescence level acquisition for PCR amplification were performed in a total volume of 25 µL, including 12.5 µL of 2 × HRM PCR Master Mix, 1.75 µL of primers mix (10 µM), 9.75 µL of RNase-Free Water, and 1 µL of template DNA (1 ng/µL). Eva Green^TM^ (Qiagen, Mainz, Germany) was utilized to monitor the accumulation of amplified product during the real-time PCR process and high-resolution melting. The melting conditions set on the Rotor-Gene Q Series software (Qiagen) and the PCR parameters were set as follows: an initial denaturation at 95 °C for 5 min, then denaturation at 9 °C for 10 s, annealing at 55 °C for 30 s, and extending at 72 °C for 24 s. This cycle was repeated 45 times. Then, amplicons were melted from 65 to 95 °C, rising by increments of 0.1 °C, and waiting for 2 s before increasing. Melt temperature peaks were visualized under green through the Rotor-Gene Q Series software V2.1.0 by computing the negative first derivative of collected fluorescence values (−dF/dT), which were plotted in their inverse format.

### 2.5. Reproductivity, Sensitivity, and Mixture Testing

Five collections of *Psilocybe cubensis* were used to examine the reproductivity of the HRM assay. Blind trials were also performed in triplicate in another accredited laboratory to avoid random effects of our HRM assay. Further, 6 DNA dilutions of *Psilocybe cubensis* from 2.0 ng to 62.5 pg (i.e., 2.0 ng, 1.0 ng, 500 pg, 250 pg, 125 pg, 62.5 pg) were prepared for the sensitivity test. The ITS and its two sub-regions ITS1 and ITS2 were all tested with real-time PCR HRM assay in these samples.

Furthermore, we mixed the genomic DNA of hallucinogenic plant *Cannabis sativa* with *Psilocybe cubensis* to produce mixture samples and analyzed the mixtures in different concentrations. The mixture test was performed on five mixtures: 90/10%, 70/30%, 50/50%, 30/70%, and 10/90% (*w*/*w*) of *Psilocybe cubensis* versus *Cannabis sativa* and a final concentration of 1 ng/µL. The mixture samples were analyzed via HRM using the primers ITS4 and ITS5. The master mix parameters were maintained for all reactions and the melt conditions remained standard with each reaction.

### 2.6. ITS Sequencing and Data Analysis

The ITS regions of *Psilocybe cubensis* and other mushrooms were directly sequenced using primers ITS4 and ITS5 to validate the taxonomic identity of our samples. The results of ITS sequencing were submitted to BLAST for cross-species sequence alignment.

The MEGA-X version 10.0.2 (Win64) was utilized to analyze the ITS, ITS1, and ITS2 sequences from each sample. A phylogenetic tree was constructed using the neighbor-joining method and the evolutionary distances were computed using the maximum composite likelihood method.

### 2.7. Unknown Sample Detection

A sample of suspected hallucinogenic mushrooms was recruited from the case for analysis. The genomic DNA of the suspected sample was extracted by the DNeasy Plant Pro Kit and diluted to 1.0 ng/µL for HRM analysis. The master mix parameters were maintained for the reaction and the melt conditions remained standard with the reaction. The three ITS regions of the suspected sample were analyzed in triplicate. To verify the results, the ITS sequence of the unknown sample was sequenced and BLAST in NCBI.

## 3. Results

### 3.1. HRM Specificity Studies

The HRM curve genetic assay could differentiate the DNA of species since each peak represented a unique amplicon from a specific species. This assay was performed to investigate whether a variation in the ITS region of different species was discriminative in derivative and normalized melting curves. The HRM profiles of the ITS, ITS1, and ITS2 amplicons of all 36 species are shown in Appendix A, and the observed melting temperatures (Tm) are summarized in Figure 1. The Tm values for fragments ITS, ITS1, and ITS2 of *Psilocybe cubensis* were 83.72 ± 0.01, 80.98 ± 0.06, and 83.46 ± 0.08 °C, respectively. Based on a combination of three Tm values and melting peaks, *Psilocybe cubensis* could be easily distinguished from the other species. Some species also presented species-specific Tm values, which are summarized in Table 2. Apparent discrepancies were also observed between some genetically close species, such as *Psilocybe cubensis* (ITS, Tm: 83.72 °C) and *Psilocybe merdaria* (ITS, Tm: 82.35 °C), as well as *Panaeolus antillarum* (ITS, Tm: 82.95 °C) and *Panaeolus papilionaceus* (ITS, Tm: 83.17 °C).

Some genetically close species or far-related species were selected to evaluate the specificity of the three ITS amplicons. As shown in Figure 2A, there are significant discrepancies in the HRM melting curves between two species from the genus *Psilocybe*. Although similar melt peaks of *Clitocybe fragrans* and *Clitocybe phyllophila* were detected in the ITS and ITS2 melt charts, they could still be discriminated against by ITS1 (Figure 2B). In addition, the HRM curve peaks of *Flammulina velutipes* were significantly different from *Clitocybe fragrans* and *Clitocybe phyllophila* even though they belonged to the same *Tricholomataceae* family. Finally, five species from different families were selected in Figure 3 to show the divergences between far-related species. Significant differences in HRM peaks were observed among almost all five species. Although the melt curves of some species were clustered together in one of the three melt charts, they could be distinguished in the other two charts. The results presented here demonstrated that the ITS HRM assay could be utilized to distinguish mushroom species. However, there were still many species that overlapped with Tm values even if based on a combination of the three ITS regions. For these species, our ITS HRM assay could only serve as a preliminary experiment before accurate species identification using ITS sequencing.

### 3.2. Reproductivity, Sensitivity, and Mixture Tests

To validate the discriminative performance of the HRM assay, the reproductivity, sensitivity, and capacity to detect species within mixture samples were tested in our study. Firstly, blind trials were performed in another accredited laboratory, and the observed HRM curves and Tm values were similar to our results (Appendix A). Further, as is shown in Figure 4A, five collections of *Psilocybe cubensis* were tested using HRM, with the same melting profiles detected in all collections. This significantly suggested the concordance of the HRM assay for the same species. In the sensitivity study, a series of dilutions of *Psilocybe cubensis* DNA from 2.0 ng to 62.5 pg was amplified by the three primer sets to determine the upper and lower limits of the HRM assays. As is shown in Figure 4B, the HRM melting curves of all three ITS amplicons presented similar peak heights when the DNA concentration ranged from 2.0 ng down to 125 pg. When the concentration was down to 62.5 pg, the melting peak heights were obviously lower than the others. Finally, a duplex real-time PCR HRM assay was performed on mixture samples that contained different ratios of *Psilocybe cubensis* and *Cannabis sativa* (Figure 5). Each duplex mix yielded two different peaks in the HRM peak chart, indicating that both species were detectable and identifiable. Furthermore, two different peaks closely related to the amount of *Psilocybe cubensis* and *Cannabis sativa* in the mixture. The average melt curve peak of the ITS region of *Psilocybe cubensis* was found at 83.10 ± 0.05 °C, while the melt curve peak of *Cannabis sativa* was found at 88.90 ± 0.15 °C in mixtures.

### 3.3. Species Validation

To confirm the reliability of our HRM analysis, it is important to validate the species designations of all samples tested based on ITS sequencing. The sequencing output showed that the length of the ITS sequence was about 620~680 bp, and the samples from the same species exhibited the same results. The obtained sequences were then compared with the sequences available in the National Center for Biotechnology Information (NCBI) internet database (GenBank). The results of the sequence alignment are summarized in Appendix A. The percent identity represents the similarity between the sequence used for alignment and the target sequence in GenBank, and we would characterize the species with the highest sequence similarity as the species of our sample. The ITS sequences of *Psilocybe cubensis* showed high sequence similarity with *Psilocybe cubensis* (KU640170), with a percent identity of 99.70%. Other species also showed similar results, with a percent identity of more than 98%, which demonstrated the identities of our samples and confirmed the credibility of our HRM data.

### 3.4. Phylogenetic Analysis

Based on the output of ITS sequencing, the nucleotide sequences were aligned using MEGA-X software, and 92.19% of the sequence within the ITS regions was observed to be variable between 36 kinds of mushrooms, which showed high polymorphism. Then, a phylogenic tree was constructed utilizing the neighbor-joining distance algorithm for the ITS region. The bootstrap consensus tree inferred from 1000 replicates was taken to represent the evolutionary history of the taxa analyzed and values greater than 50% were indicated on the branches. As is shown in Figure 6, the phylogenic tree generated by ITS sequences separated different species into different branches, which clearly identified the differences in ITS sequences between species. The two species from the genus *Clitocybe* were gathered into the same terminal branches, while *Psilocybe cubensis* and *Psilocybe merdaria* were in separate branches, which was consistent with our HRM data shown in Figure 2. Although *Flammulina velutipes* belonged to the same family as the two *Clitocybe* species, they showed genetic distances in our phylogenic tree, which interpreted the divergences in the HRM curves between these three species. Furthermore, as expected, the five species from different families in Figure 3 were mostly separated into different branches, except *Lepista sordida* and *Clitocybe fragrans* that showed relatively close genetic distances. Our phylogenic tree further supported the HRM data presented above and verified the reliability of our study.

### 3.5. Unknown Sample Detection

The genomic DNA of the suspected sample was extracted successfully and diluted to 1.0 ng/µL. In the HRM analysis, the results showed that the Tm values of the sample’s ITS, ITS1, and ITS2 were 83.88 ± 0.12, 81.03 ± 0.08, and 83.44 ± 0.05 °C, which perfectly matched the HRM curves and Tm values of *Psilocybe cubensis* in our research (Figure 7). In addition, the sequencing results also matched to *Psilocybe cubensis* (KU640170, 99.71%). This suspected sample was therefore identified as *Psilocybe cubensis*.

## 4. Discussion

The data we present here demonstrate that the HRM assay allowed for the discrimination of 36 mushroom species based on the ITS, ITS1, and ITS2 regions, even if in the mixture samples. The ITS sequencing performed for species validation further interpreted our HRM data and verified the reliability of our study. This study proved the feasibility of the PCR HRM assay to discriminate *Psilocybe cubensis* and other fungi in cases and provided evidence of establishing a valuable database containing HRM experimental analysis data of hallucinogenic mushrooms and plants.

As we previously discussed in the Introduction, HRM analysis is a reliable and sensitive method for the discrimination of DNA barcoding. We therefore sought to establish a fungi discrimination method based on ITS HRM analysis. However, insufficient variation in nucleotides inside the ITS region may still cause similar Tm values and melting curves due to the slight discrepancies in ITS sequences among some closely related species [25,26,27,28]. In our study, to detect nucleotide variations more accurately, the saturating dye Eva Green^TM^ was used in our HRM analysis. The excitation and emission spectra of Eva Green^TM^ were very close to those of fluorescein, with the advantage of no PCR inhibition, strong combinatorial ability, and high GC analysis accuracy [29]. Moreover, three primer sets were used to amplify the ITS region and its highly polymorphic sub-regions ITS1 and ITS2. These three amplicons were all analyzed by HRM to improve the discrimination power of our HRM curve genetic assay. To avoid random effects derived from the environment, personnel operation, and instruments, we conducted each test in triplicate on two different instruments and obtained the standard deviations of Tm values. However, error bars were not consistent across the same primer set due to different sample purities as well as different binding abilities of primers with each sample. Based on the combination of melting temperatures and curves obtained for the three amplicons, *Psilocybe cubensis* and some species such as *Clitopilus crispus* and *Lanmaoa asiatica* were successfully distinguished from others. For each of the above species, the three ITS amplicons performed differently, which might interpret the obvious species-specific results obtained from these species. Several species were also selected to assess the specificity of the ITS HRM assay (Figure 2 and Figure 3). The chosen species were clearly discriminated by the HRM melting peaks and Tm values, regardless of species from the same genus or different families. Even though some species could not be distinguished by one of the three amplicons, they could be distinguished by the other two. These results clearly demonstrate that the discrimination performance of the ITS could be supplemented by its two sub-regions, ITS1 and ITS2 (Figure 2). However, there are still some species with overlapping Tm values, which might be attributed to similar ITS sequences derived from genetically close species. In addition, some ITS amplicons with dissimilar sequences may also accidently create overlapping Tm values. Although we sought to improve the accuracy of the HRM assay, the Tm difference of less than 0.5 °C could not be distinguished due to random effects. For these species, our ITS HRM assay only served as a preliminary experiment before ITS sequencing. Yet for the species presenting species-specific results (Table 2), our ITS HRM assay can clearly distinguish them, and it decreased the species differentiation time. Additionally, the ITS region was directly sequenced, and the sequences of all samples were aligned to construct a phylogenic tree (Figure 6). Intriguingly, some species from the same genus did not appear on the same terminal branches, suggesting that morphologically defined species were not supported by the ITS sequence, which was in accordance with the study of Nugent et al. [10]. The species which appeared to have significant differences in the HRM assay almost appeared on separate branches, such as *Psilocybe cubensis* and *Psilocybe merdaria*, or *Flammulina velutipes* and the other two species from *Tricholomataceae*. The species that produced similar HRM melt peaks also appeared on the same terminal branch, such as the two *Clitocybe* species. These all indicate the consistency of our data obtained from HRM analysis and sequencing and validate the reliability of our ITS HRM assay.

The sensitivity for the ITS HRM assay was also evaluated using a series of DNA dilutions of *Psilocybe cubensis* ranging from 2.0 ng to 62.5 pg. Except for the DNA input quantity of 62.5 pg with a relatively low peak height, all DNA dilutions produced excellent melt peaks, which significantly demonstrate that HRM is a highly sensitive approach. Furthermore, the HRM analysis was performed on DNA mixtures of *Psilocybe cubensis* and *Cannabis sativa*. The results acquired from the tests demonstrate that HRM analysis could be utilized to assess mixtures of species, even though the DNA mix of two species is not equilibrated (Figure 5). Interestingly, the HRM peak heights were obviously affected by the DNA concentrations in mixtures, though the DNA input quantities in all mixtures were higher than 100 pg. This might be attributed to the competition of different DNAs during the amplification process. Mixtures of illegal hallucinogenic mushrooms and plants are commonly encountered during forensic investigations. Although the sensitivity of the HRM assay might be affected in mixtures, the ratio of HRM peak heights can determine the amount of each contributor within mixture samples, which is certainly practical for forensic investigations.

The identification of fungi at the species level is critical for many research areas, such as health sciences, agriculture, and forensic science, and for determining whether hallucinogenic mushrooms are essential for treatment or are being trafficked as drugs [13,16,22]. In addition, studies have shown that psilocybin may relieve depression and so this HRM method could also be utilized for research in this area [30]. In this study, HRM analysis based on DNA barcoding the ITS made a clear distinction between *Psilocybe cubensis* and the other 35 mushrooms. Additionally, the reproductivity, sensitivity, and mixture tests all demonstrated that the ITS HRM analysis is a highly sensitive and practical approach. The limitation of the current research is due to the lack of samples. Further studies with larger sample sizes are needed to guarantee the probability of the correct identification and enhance the species diversity of the HRM profile database. Another issue is that some species presented overlapping Tm values. Therefore, supplementary barcodes were required for more accurate taxonomic studies, such as the largest and second largest subunit of RNA polymerase II (RPB1 and RPB2) [12,31,32]. Despite these limitations, our study still verifies that the HRM assay has the potential to become a powerful molecular tool for forensic mushroom taxonomic studies, especially when it comes to tracking the distribution network and associating evidence for criminal cases. Authors should discuss the results and how they can be interpreted from the perspective of previous studies and the working hypotheses. The findings and their implications should be discussed in the broadest context possible. Future research directions may also be highlighted.

## 5. Conclusions

In conclusion, the method of HRM analysis can quickly distinguish the DNA barcoding of *Psilocybe cubensis*, which can be utilized for drug-related cases and forensic science. This method can provide data for the preliminary construction of DNA barcoding database of hallucinogenic mushrooms and plants, and develop a rapid and scientific method for species identification. 

## Figures and Tables

**Figure 1 genes-12-00199-f001:**
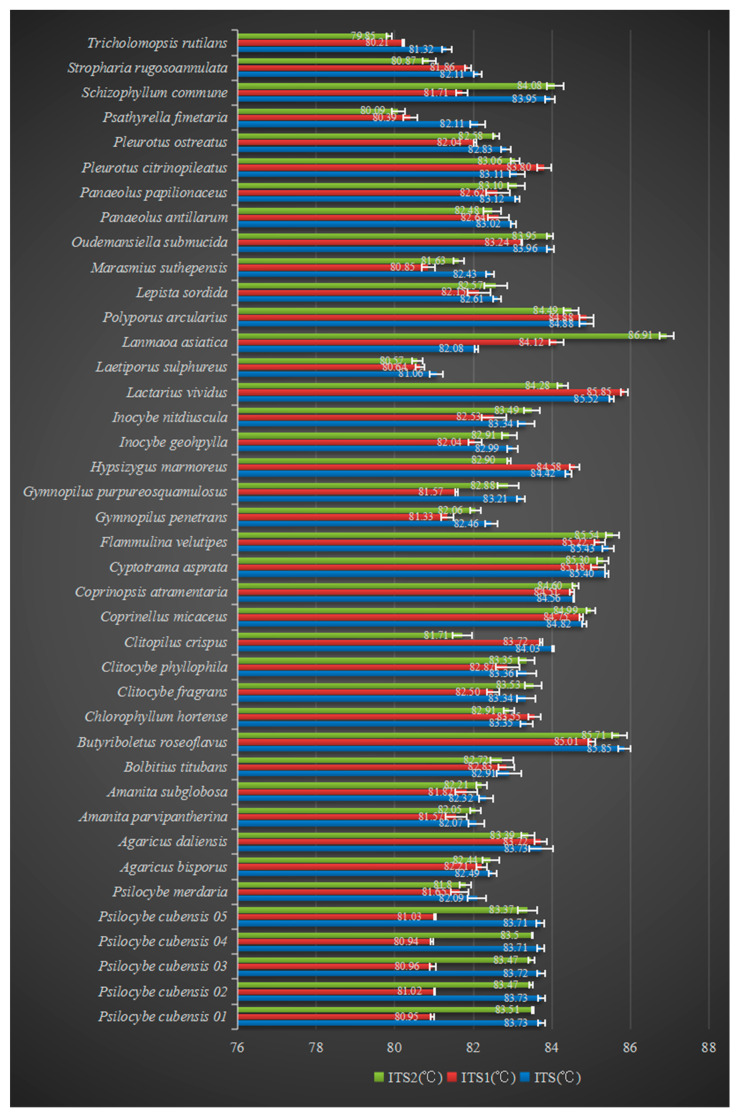
High-resolution melting (HRM) melting temperatures (Tm) of ITS, ITS1, and ITS2 of 40 mushrooms.

**Figure 2 genes-12-00199-f002:**
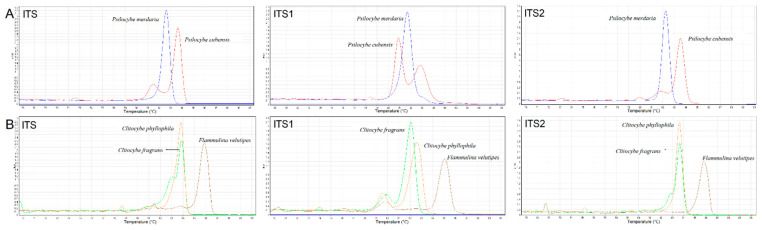
Specificity of ITS, ITS1, and ITS2 regions between genetically close species (**A**): *Psilocybe cubensis* and *Psilocybe merdaria*; (**B**): *Clitocybe fragrans*, *Clitocybe phyllophia* and *Flammulina velutipes*). The negative first derivative (−dF/dT) of the normalization melt curve from HRM.

**Figure 3 genes-12-00199-f003:**
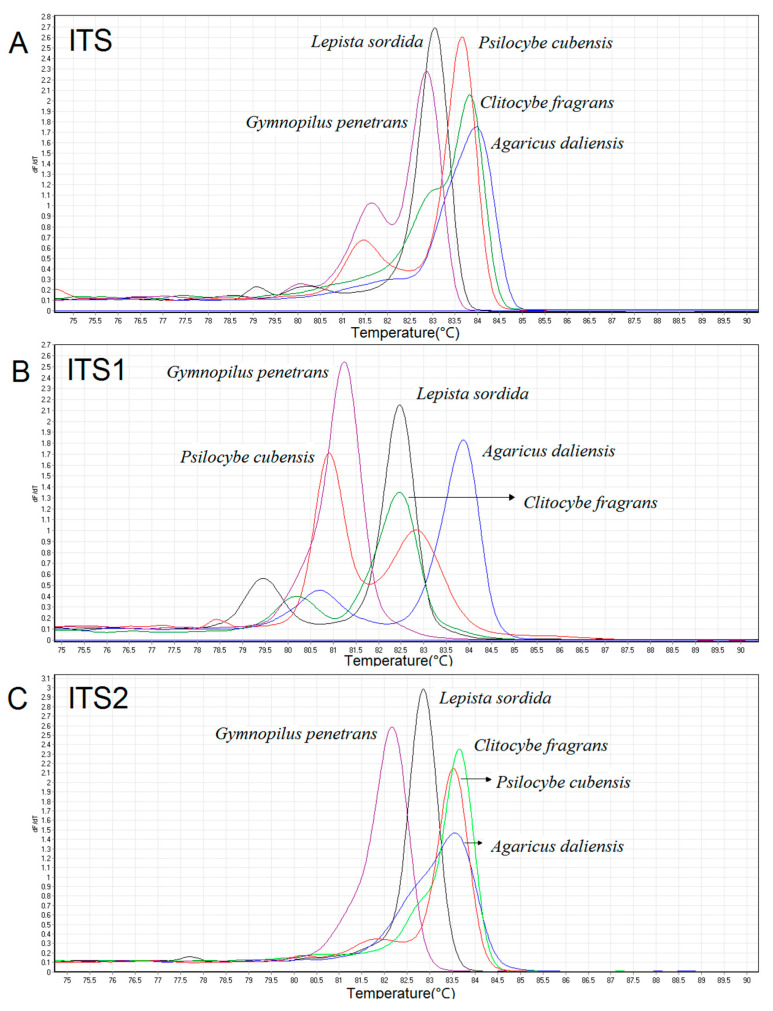
Specificity of ITS (**A**), ITS1 (**B**), and ITS2 (**C**) regions between species from different families. The negative first derivative (−dF/dT) of the normalization melt curve from HRM (*Gymnopilus penetrans*: purple curves; *Lepista sordida*: black curves; *Clitocybe fragrans*: green curves; *Psilocybe cubensis*: red curves; *Agaricus daliensis*: blue curves).

**Figure 4 genes-12-00199-f004:**
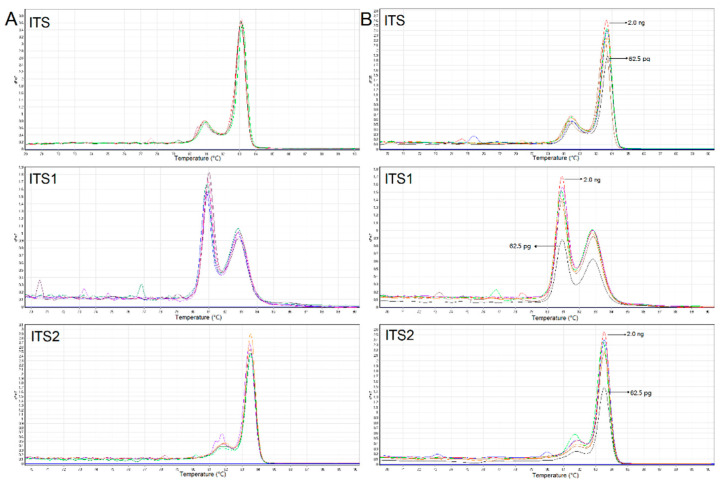
(**A**) HRM profile of ITS, ITS1, and ITS2 amplicons from five *Psilocybe cubensis* samples. (**B**) HRM profile of *Psilocybe cubensis* DNA ranging from 2.0 ng to 62.5 pg for the sensitivity study. The melt curve was plotted as the first derivative (dF/dT) versus temperature (°C).

**Figure 5 genes-12-00199-f005:**
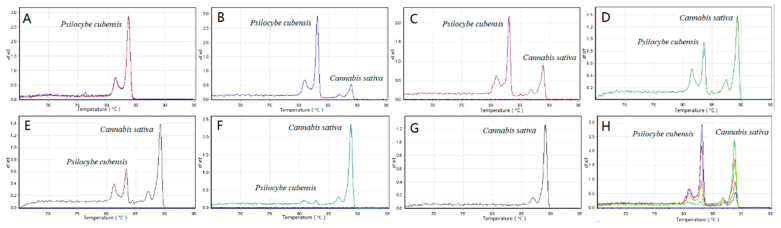
Variation in the HRM profile of the ITS region. The negative first derivative (−dF/dT) of the normalization melt curve of rDNA ITS from HRM. Mixtures includes different contents of *Psilocybe cubensis* and *Cannabis sativa*: (**A**) 100/0%, (**B**) 90/10%, (**C**) 70/30%, (**D**) 50/50%, (**E**) 30/70%, (**F**) 10/90%, (**G**) 0/100%, and (**H**) is a summary of all situations.

**Figure 6 genes-12-00199-f006:**
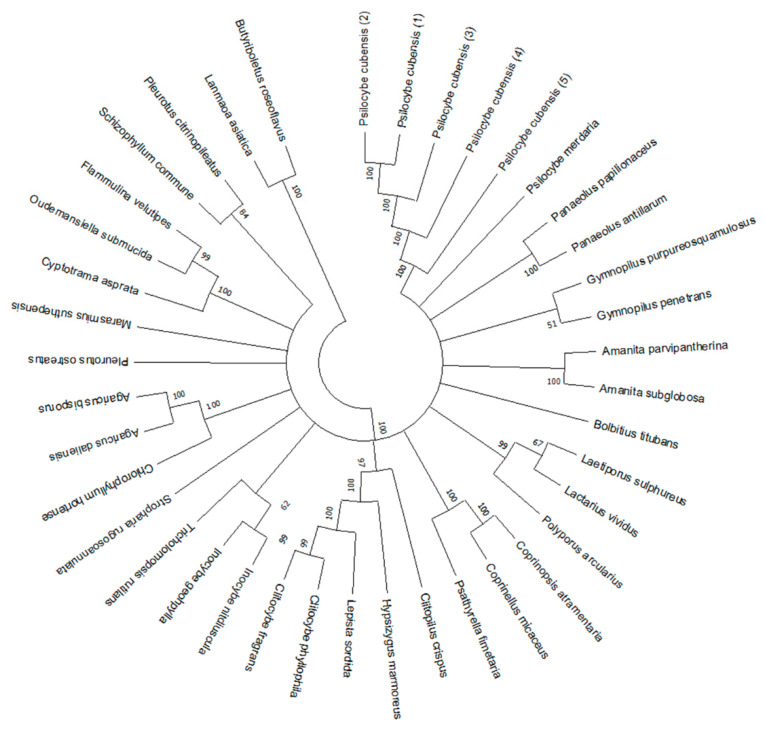
The phylogenetic tree was constructed using the neighbor-joining method and operated by MEGA-X-10.0.2 software. The evolutionary distances were computed using the maximum composite likelihood method. Analysis was based on 40 aligned sequences from the ITS locus.

**Figure 7 genes-12-00199-f007:**
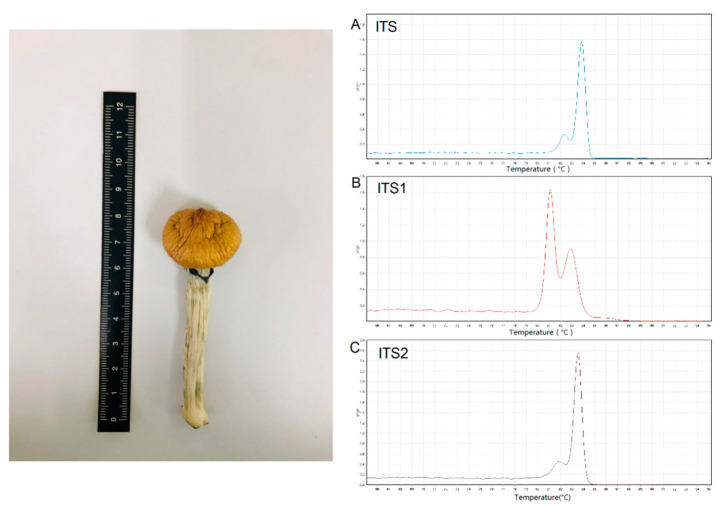
HRM profile of the ITS (**A**), ITS1 (**B**), and ITS2 (**C**) amplicons from the unknown sample.

**Table 1 genes-12-00199-t001:** Primers for amplifying DNA barcoding of *Psilocybe cubensis* and other species.

Scheme 5	Primer	Sequences (5′→3′)	Gene	Tm (°C)	Size of Fragments
ITS	ITS4	TCCTCCGCTTATTGATATGC	ITS1-5.8S-ITS2	51.7	~680 bp
	ITS5	GGAAGTAAAAGTCGTAACAAGG		50.7	
ITS1	ITS2	GCTGCGTTCTTCATCGATGC	ITS1	57.4	~275 bp
	ITS5	GGAAGTAAAAGTCGTAACAAGG		50.7	
ITS2	ITS4	TCCTCCGCTTATTGATATGC	ITS2	51.7	~310 bp
	ITS86	GTGAATCATCGAATCTTTGAAC		48.9	

**Table 2 genes-12-00199-t002:** Mushrooms that can be discriminated against based on Tm values of all three regions.

Species	ITS (°C)	ITS1 (°C)	ITS2 (°C)
*Psilocybe cubensis*	83.72 ± 0.01	80.98 ± 0.06	83.46 ± 0.08
*Agaricus daliensis*	83.73 ± 0.30	83.72 ± 0.16	83.39 ± 0.17
*Clitopilus crispus*	84.03 ± 0.03	83.72 ± 0.04	81.71 ± 0.25
*Gymnopilus purpureosquamulosus*	83.21 ± 0.11	81.57 ± 0.04	82.88 ± 0.27
*Hypsizygus marmoreus*	84.42 ± 0.08	84.58 ± 0.13	82.90 ± 0.05
*Lactarius vividus*	85.52 ± 0.05	85.85 ± 0.09	84.28 ± 0.13
*Laetiporus sulphureus*	81.06 ± 0.17	80.64 ± 0.11	80.57 ± 0.14
*Lanmaoa asiatica*	82.08 ± 0.05	84.12 ± 0.18	86.91 ± 0.18
*Marasmius suthepensis*	82.43 ± 0.10	80.85 ± 0.17	81.63 ± 0.14
*Oudemansiella submucida*	83.96 ± 0.08	83.24 ± 0.01	83.95 ± 0.07
*Pleurotus ostreatus*	82.83 ± 0.12	82.04 ± 0.03	82.58 ± 0.08
*Psathyrella fimetaria*	82.11 ± 0.19	80.39 ± 0.18	80.09 ± 0.17
*Schizophyllum commune*	83.95 ± 0.12	81.71 ± 0.14	84.08 ± 0.21
*Stropharia rugosoannulata*	82.11 ± 0.09	81.86 ± 0.08	80.87 ± 0.17
*Tricholomopsis rutilans*	81.32 ± 0.12	80.21 ± 0.02	79.85 ± 0.07

## Data Availability

The data presented in this study are available in Figure 1, Appendix A.

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
