# Peer review of "A Forensic Detection Method for Hallucinogenic Mushrooms via High-Resolution Melting (HRM) Analysis"

_genes, 2021, doi:10.3390/genes12020199_

Round 1
Reviewer 1 Report
I found the article very original and with scientific soundness and well written.
It is an interesting different approach to identify psychoactive mushrooms, but the application could not be limited to forensic sciences.
The main question addressed by the research, since the technique allowed to identify hallucinogenic mushrooms of the genus psilocybe for national drug control legislation. Moreover, I found this article useful for identify Psilocybe species, if these mushrooms will be further used to extract psilocybin for the treatment of depression as some studies propose.
Therefore, I only suggest that the authors provided further fields of application, some of them besides forensic sciences.
Author Response
Reviewer #1: I found the article very original and with scientific soundness and well written. It is an interesting different approach to identify psychoactive mushrooms, but the application could not be limited to forensic sciences. The main question addressed by the research, since the technique allowed to identify hallucinogenic mushrooms of the genus psilocybe for national drug control legislation. Moreover, I found this article useful for identify Psilocybe species, if these mushrooms will be further used to extract psilocybin for the treatment of depression as some studies propose. Therefore, I only suggest that the authors provided further fields of application, some of them besides forensic sciences.
Reply: We appreciated for the reviewer’s suggestion. As the reviewer said, psilocybin has a positive effect on the depression and some progress has been made. As suggested, the related references have been added in our discussion (line 322-324).
Reviewer 2 Report
Review genes; Forensic Detection Method …
Generally:
Where are the sequences that were checked in NCBI/BLAST? They should be listed in a supplemental
When the authors calculated the “identification” and gave a probability match, how was this done?
Why when some of the samples had multiple peaks were the secondary peaks not included for better discrimination?
Although I think I know why, the authors should explain why the error bars are not consistent across the same primer set.
Lastly, the discussion rambles a bit exploring alternatives and making assertions reporting other work. If this is background materials it belongs in introduction or they need to tie these discussion points to the overall goals of the paper. The discussion reads more like a literature review than a discussion of the acquired data.
Page 1; line 33 looking at the word “addictive” with respect to this reference, the reference does not indicate any addictive properties to these compounds. This word needs to be changed.
Page 3; line 101 never heard of a Rotor-Gene Q “SERIOUS”. I have a Rotor-Gene Q and I looked it up on-line as well and found nothing under serious. Is this a misprint? Maybe they mean series?
Page (the page numbers are incorrect; after page 7 of 15 it changes to 1 of 15. This is the page where Figure 3 is.
The page 1 of 15 with figure 3; line 178 it would be very helpful to the researcher and reader to place these blind trial results in a supplemental form.
Page 3 of 15 I believe the Reproducibility, sensitivity, and mixture testing OUTLINE is certainly adequate and follows the guidelines, in the USA, of SWGDAM (https://www.swgdam.org/publications)
Page 4 of 15 In the verification of results the authors used only one sample it seems. It might be better for the authors to use several knowns to further bolster their results. The statistical “magic number” five might be reasonable.
Page 5 of 15 I think they present their results in a fair way in visual form giving the reader an overall look at the differences between species, however, it would be much better to put these data in a matrix with each specials on the abscissa and the ordinate and then compare differences between them taking into account the error bars. This then would allow the reader to know how many differences between samples there are. For instance the matrix would show the number of differences between each sample and thus the reader would be informed as to how good a match might be.
Page 6 of 15 Table 2 gives the reader raw data, however, I think this needs a matrix as well. They use the words “significantly identified’ and they do NOT explain what this means. Identification in forensics MOST ALWAYS uses some kind of statistic to inform the jury or investigator how important or significant the match is. This is done by Bayesian stats most times in forensics so these authors need to base this statement on some statistical calculation for the reader.
Page 6 of 15 line 165, they again use the word “identify”. See my paragraph above.
This is where the pages go wrong. They become 1 of 15 and have Figure 3 on the page. Figure 3 is complex with many curves but it is hard to see which peak belongs to which sample. This figure needs to be in color OR different kinds of lines or dots making up a curve so the reader can discern which goes where.
Second page 1 of 15 Line 179 What does “consistent of our results mean? Again; how many blind trial samples were there?
Second page 2 of 15 Figure 4 and Figure 5 present data which is commensurate with forensic guidelines which are nearly universal. These data presented in this form is excellent and shows reproducibility and mixture analysis.
Second page 2 of 15 Line 207 I know this seems like a large sequence that can be used for HRM but sequencing as large as this have been published in the last few years using this large a sequence with good results. Just a remark for the authors.
Second Page 3 of 15 Line 216 This sentence states 90% of variable sites. The authors need to explain this more. Not sure what this actually means, sequence based or some other metric?
Second page 6 of 15 Line 318 Again “identification” is used too liberally here. See my comments before on this concept and word. A specific figure or table needs to be cited here.
Second page 6 of 15 IMPORTANT; Supplemental Figures need a better description of the sample numbers and what they mean and or why they are marked the way they are. Is there a meaning to the letters and numbers?
The authors clearly need to put more work into their manuscript such as the use of the word "identify". This is a important concept in a court of law as well as for other forensic scientists. They need to show how they come up with this word and they need to prove to the reader that this is true. This is done by a statistical weight. If the authors wish to use these data as an "investigative lead" than that is different. No stats would be needed but it would be still desirable.
Author Response
Reviewer #2:
Firstly, our manuscript has undergone English language editing by MDPI. The following are the response to reviewer’s comments.
Where are the sequences that were checked in NCBI/BLAST? They should be listed in a supplemental
Reply: We would like to thank the reviewer for the rigorous attitude. As suggested, we have listed all sequencing results in supplementary Table 1.
When the authors calculated the “identification” and gave a probability match, how was this done?
Reply: The percent identity in sequence alignment results in NCBI represented the similarity between the sequencing output and target sequence in GenBank. According to the BLAST results, we would characterize the species with highest sequence similarity as the species of our sample. As suggested, identification standard have been added in line 225-227.
Why when some of the samples had multiple peaks were the secondary peaks not included for better discrimination?
Reply: We appreciate for the reviewer’s remind. Just as the reviewer said, there were some samples with secondary peaks. The reason is that we used universal primers for the amplification of ITS sequence in all samples, which may lack species specificity in some samples. It is highly possible that there is nonspecific amplification in these samples, where sequences outside the target region are amplified. But the existence of secondary peaks is also species-specific, which improves the specificity of the HRM curves. Since the secondary peaks may not be obtained from target region, we didn’t add their Tm values in our table. But they could be observed in supplementary figures, which were provided as a supplementary evidence for fungi identification.
Although I think I know why, the authors should explain why the error bars are not consistent across the same primer set.
Reply: The inconsistent error bars may derive from the different sample purity. Further, the binding ability of universal primers with different sample may be also different (line 279-283). These all make the error bars inconsistently.
Lastly, the discussion rambles a bit exploring alternatives and making assertions reporting other work. If this is background materials it belongs in introduction or they need to tie these discussion points to the overall goals of the paper. The discussion reads more like a literature review than a discussion of the acquired data.
The discussion section need amended appropriately.
Reply: We are grateful for the reviewer’s suggestions. As suggested, we have made some modifications to the discussion section of the article, and the discussion of several other methods and the reason we choose HRM have been removed into the introduction (line 62-69, 71, 77-80, 269-271).
Page 1; line 33 looking at the word “addictive” with respect to this reference, the reference does not indicate any addictive properties to these compounds. This word needs to be changed.
Reply: We appreciated for the reviewer’s suggestion. The word “addictive” has been changed to “abuse and hazardous to health” (line 34).
Page 3; line 101 never heard of a Rotor-Gene Q “SERIOUS”. I have a Rotor-Gene Q and I looked it up on-line as well and found nothing under serious. Is this a misprint? Maybe they mean series?
Page (the page numbers are incorrect; after page 7 of 15 it changes to 1 of 15. This is the page where Figure 3 is.
Reply: We are so sorry for making mistakes. The instrument we used is Rotor-Gene Q and the parameter settings is performed on the Rotor-Gene Q Series Software (Qiagen), rather than “serious” (line 116). The wrong page number has been corrected.
The page 1 of 15 with figure 3; line 178 it would be very helpful to the researcher and reader to place these blind trial results in a supplemental form.
Reply: As suggested, we provided a supplemental table (Table S2) to place the blind trial results from other laboratories.
Page 3 of 15 I believe the Reproducibility, sensitivity, and mixture testing OUTLINE is certainly adequate and follows the guidelines, in the USA, of SWGDAM (https://www.swgdam.org/publications)
Page 4 of 15 In the verification of results the authors used only one sample it seems. It might be better for the authors to use several knowns to further bolster their results. The statistical “magic number” five might be reasonable.
Reply: We would like to thank the reviewer for the rigorous attitude. The purpose of our study was to analyze the inter-species variations and establish a novel approach for identifying hallucinogenic mushrooms in forensic investigation. We therefore collected mushrooms from thirty-six species to compare the difference between melting curves from different species. Further, to validate the precision of our study, we prepared five different individuals of Psilocybe cubensis, which is the most common hallucinogenic mushroom in forensic investigation. Five completely overlapping curves were obtained from these five samples, which suggested that different individuals from same species would produce same HRM curves and demonstrated the reliability of our assay. We have also tested all samples in triplicate to ensure the repeatability and reliability of the assay. However, unlike Psilocybe cubensis, samples from other thirty-five species were hard to recruit. Psilocybe cubensis is common in drug trafficking cases and could be directly collected from our forensic labs, while other mushrooms, especially hallucinogenic and toxic mushrooms, were rare or under strict control in China. For these reasons, we are unable to collect more samples from same species and further verify the reproducibility of other species.
Page 5 of 15 I think they present their results in a fair way in visual form giving the reader an overall look at the differences between species, however, it would be much better to put these data in a matrix with each specials on the abscissa and the ordinate and then compare differences between them taking into account the error bars. This then would allow the reader to know how many differences between samples there are. For instance the matrix would show the number of differences between each sample and thus the reader would be informed as to how good a match might be.
The authors clearly need to put more work into their manuscript such as the use of the word "identify". This is an important concept in a court of law as well as for other forensic scientists. They need to show how they come up with this word and they need to prove to the reader that this is true. This is done by a statistical weight. If the authors wish to use these data as an "investigative lead" than that is different. No stats would be needed but it would be still desirable.
Reply: We appreciated for the reviewer’s suggestion. Combining with some statistical methods would make the experimental results more convincing. However, the purpose of our study is to provide a reference for rapid identification of Psilocybe cubensis by HRM curves and Tm. These data are only used as an investigative lead. Therefore, these statistical methods are not essential in our study. But these statistical methods should be performed if someone will apply our results as a scientific evidence in the jury.
Page 6 of 15 Table 2 gives the reader raw data, however, I think this needs a matrix as well. They use the words “significantly identified’ and they do NOT explain what this means. Identification in forensics MOST ALWAYS uses some kind of statistic to inform the jury or investigator how important or significant the match is. This is done by Bayesian stats most times in forensics so these authors need to base this statement on some statistical calculation for the reader.
Reply: We would like to thank the reviewer for the rigorous attitude. The “significantly identified” means the differences in HRM curves and Tm between species. We are sorry for our inappropriate statements and have corrected our statements in Table 2. Further, the reason of not using Bayesian stats have been explained above.
Page 6 of 15 line 165, they again use the word “identify”. See my paragraph above.
Reply: we appreciated for the reviewer’s suggestion. We must admit that our data are not under statistical examination and could not be used to inform the jury. Therefore, we have corrected all the word “identify” mentioned in our results and the discussion of our results as the word “distinguish” or “discriminate”.
This is where the pages go wrong. They become 1 of 15 and have Figure 3 on the page. Figure 3 is complex with many curves but it is hard to see which peak belongs to which sample. This figure needs to be in color OR different kinds of lines or dots making up a curve so the reader can discern which goes where.
Reply: As suggested, the different HRM curves and colors represented by the species have been shown in the figure and diagram (line 186-188). Also, the page mistake has been corrected.
Second page 1 of 15 Line 179 What does “consistent of our results mean? Again; how many blind trial samples were there?
Reply: The “consistent” means that the HRM results from second laboratory are almost overlapped with our results. We have provided a supplementary table to show Tm values obtained from second laboratory as reviewer’s suggestion in supplementary table. Since the word “consistent” is inappropriate, we have corrected this word as “similar”. Further, each sample had been analyzed as blind trial sample.
Second page 2 of 15 Line 207 I know this seems like a large sequence that can be used for HRM but sequencing as large as this have been published in the last few years using this large a sequence with good results. Just a remark for the authors.
Reply: We agreed with reviewer. In previous years, the study of ITS mainly focused on ITS sequencing, which took a long time for data analysis. The ITS HRM assay in our study only costs two hours for examination and provides species-specific results. Further, there are few experiments to analyze ITS sequence with HRM, and the sample size is also limited. Our study not only analyzed ITS sequence and its two sub-regions ITS1 and ITS2, but also collected up to 36 samples, improving species diversity of existing HRM data.
Second Page 3 of 15 Line 216 This sentence states 90% of variable sites. The authors need to explain this more. Not sure what this actually means, sequence based or some other metric?
Reply: We appreciated for the reviewer’s question. As modified in the article, the “variable sites” means sequence differences between mushrooms in our study. We have corrected the sentences as “92.19% of the sequence within the ITS regions were observed variable between 36 kinds of mushrooms, which showed high polymorphism”. (line 233-234)
Second page 6 of 15 Line 318 Again “identification” is used too liberally here. See my comments before on this concept and word. A specific figure or table needs to be cited here.
Reply: We appreciated for the reviewer’s suggestion. According to reviewer’s suggestion, several “identification” had been changed to other words (line 50, 73,86-87, 291, 298, 326).
Second page 6 of 15 IMPORTANT; Supplemental Figures need a better description of the sample numbers and what they mean and or why they are marked the way they are. Is there a meaning to the letters and numbers?
Reply: We are very grateful for the reviewer's remind. The numbers in the figure are derived from the sample numbers in the supplementary table. The samples from same genus would be divided into same group, such as A group, B group and so on. Specific descriptions have been listed in line 341-347.